# Recent Progress in Research on [2.2]Paracyclophane-Based Dyes

**DOI:** 10.3390/molecules28072891

**Published:** 2023-03-23

**Authors:** Wenjing Liu, Huabin Li, Yanmin Huo, Qingxia Yao, Wenzeng Duan

**Affiliations:** 1School of Chemistry and Chemical Engineering, Shandong Provincial Key Laboratory of Chemical Energy Storage and Novel Cell Technology, Liaocheng University, Liaocheng 252000, China; 2Shandong Xinfa Ruijie New Material Co. Ltd., Liaocheng 252000, China

**Keywords:** [2.2]paracyclophane, fluorescent dye, fluorescent probe, AICPL

## Abstract

In recent years, the [2.2]paracyclophane (PCP) ring has attracted extensive attention due to its features of providing not only chirality and electron-donating ability but also steric hindrance, which reduces intermolecular π–π stacking interactions and thereby improves the fluorescence properties of dyes. To date, some circularly polarized luminescence (CPL)-active small organic molecules based on the PCP skeleton have been reviewed; however, the application of the PCP ring in improving the photophysical properties of fluorescent dyes is still limited, and new molecular design strategies are still required. This review summarizes and promotes the application of PCP in fluorescent dye design, fluorescence detection, and CPL modulation. We expect that this review will provide readers with a comprehensive understanding of the PCP skeleton and lead to further improvement in fluorescent dye design.

## 1. Introduction

[2.2]Paracyclophane (PCP) is a typical cyclophane that was first synthesized and isolated in 1949 as a pyrolysis product of para-xylene [1]. PCP includes two strongly interacting benzene unit “decks” with a distance of ~3.09 Å, which are connected by two ethylene “bridges” (~2.78 Å) (Figure 1) [2]. The two benzene rings stacked at such close proximity can lead to transannular π–π strain in the aromatic rings and cause deviation from the normal molecular structure. The strain, distorted structure and transannular effects alter the photophysical and optoelectronic properties of PCP scaffolds. Therefore, due to the intriguing structure and the unusual photophysical [3] and optoelectronic properties [4] of PCP scaffolds, they have been successfully used as privileged scaffolds in asymmetric synthesis [5], π-stacked polymers [6], energy materials [7], and organic fluorescent dyes [8] and have aroused wide interest in biological and materials science fields [9].

Conventional organic fluorescent dyes include cyanines [10], BODIPYs (4,4-difluoro-4-bora-3a,4a-diazas-indacene) [11], rhodamine analogues [12], squaraines [13], and porphyrins [14]. BODIPY dyes have a large π-conjugated skeleton and exhibit excellent photophysical properties in dilute solution. However, when in high concentration in solution or a state of aggregation, BODIPY dyes tend to adopt a face-to-face stacking mode (*H*-aggregation) and usually display an aggregation-caused emission quenching (ACQ) effect, which limits their application. Fortunately, the ACQ effect can be effectively overcome using an aggregation-induced emission (AIE) strategy, which was introduced by Tang et al. and has since been widely used [15]. Many AIE skeletons, such as 1,1,2,3,4,5-hexaphenylsilone and tetraphenylethylene, have been developed and applied in organic fluorescent dyes. These skeletons have twisted structures and can effectively inhibit the intermolecular π–π interactions occurring in the aggregation state. Taking advantage of PCP scaffolds, the introduction of a PCP group to BODIPY or BODIPY analogues can also result in elimination of the strong π–π interactions between the intermolecular indacene planes and achieve an AIE effect. Therefore, the introduction of the PCP skeleton to organic fluorescent dyes improves the photophysical properties of fluorescent dyes and has received increasing attention in recent decades.

Circularly polarized luminescence (CPL) has many potential applications in chirality sensing [16], optical displays [17,18,19], and chiroptical materials [20,21]. The two key factors to satisfy the requirements of the above applications during the development of CPL materials are high fluorescence quantum yield (*Φ*_f_) and high luminescent dissymmetry factors (|*g*_lum_|) [22]. In recent years, CPL-active small organic molecules (CPL-SOMs) have received wide attention due to their easy derivatization, excellent processability, tunable wavelengths, high fluorescence quantum yields, and low toxicity [23,24,25,26,27]. However, many CPL-SOMs suffer from the disadvantages of small |*g*_lum_| values and aggregation-caused quenching (ACQ). An AIE strategy is also effective for achieving higher *Φ*_f_ and larger |*g*_lum_| values, and has been applied to overcome ACQ in CPL-SOMs [28]. It is well known that AIE-active SOMs display no or weak fluorescence in dilute solution but become highly emissive as aggregates or in a solid state, which may increase *Φ*_f_ and amplify the |*g*_lum_| values. Because of the unique chemical structure, specific photophysical properties, and stable planar chirality of the PCP, AIE-active SOMs based on a PCP skeleton are receiving increasing attention. Recently, we reported on the design and synthesis of AIE-active PCP derivatives [29,30], which exhibit aggregation-induced circularly polarized luminescence (AICPL) signals with moderate *Φ*_f_ and |*g*_lum_| values. Therefore, in this mini review, we focus on providing an overview of the development of PCP skeletons for application in fluorescent dye design, fluorescence detection, and CPL modulation in order to facilitate their future application in chiral fluorescent dyes.

## 2. Application of PCP Skeleton in Organic Fluorescent Dyes

### 2.1. Application of PCP Skeleton in Modifying Dyes

It is well known that many reported organic fluorescent dyes suffer from small Stokes shifts, which affects their fluorescence emission efficiency. Therefore, it is necessary to develop a fluorescent dye with large Stokes shifts, superior optical properties, and good stability [31]. Due to the special electronic structure, [2.2]paracyclophanyl fluorescent dyes exhibit intrinsic fluorescence and can be applied to access new fluorescent dyes [32]. In 2019, Delcourt’s group reported a series of [2.2]paracyclophane-fused coumarin systems (Figure 1) [33], which are compact heteroaromatic PCP dyes with unique three-dimensional (3D) structures. The introduction of the PCP skeleton allows the synthesis of 3D coumarin systems and improvement of the photophysical properties through large Stokes shifts and red-shifted absorption and emission bands. Furthermore, when planar chiral 4-formyl [2.2]paracyclophane was introduced to these [2.2]paracyclophane-fused coumarins, the resulting product exhibited promising chiroptical properties with |*g*_lum_| up to 5 × 10^−3^.

In order to investigate the influence of the PCP motif and its “phane” interaction on the spectroscopic properties of fluorophores 1−5, the authors compared the different photophysical properties of these emitters (Table 1). Compared with [2.2]paracyclophane (Table 1, entry 1), the 3D coumarins (Table 1, entries 2−6) exhibited red-shifted absorption and emission bands. Moreover, when compared with the absorption and emission maxima of 4-amino [2.2]paracyclophane (Table 1, entry 7) and paracyclophane-deprived model compound 5a (Table 1, entry 9), the 3D coumarins also showed red-shifted absorption and emission bands. These results indicate that larger π-conjugation systems form between the PCP skeleton and the coumarin skeleton. Importantly, compared with commercially available coumarin 4, the wavelength absorption in the UV−vis spectra of fluorophores 1−3 was lower, and a remarkably large Stokes shift (230 nm) was obtained for coumarin 2d. In addition, in comparison with the behaviors of their unsubstituted analogues (±)-1b and (±)-1d, the 3D coumarins (±)-2a and (±)-2b with bromine atoms showed observed hypsochromic shifts of the emission bands, which indicated that the through-space interactions in the PCP skeleton could influence the photophysical properties of these 3D coumarins. Therefore, these results demonstrate that PCP can be introduced into coumarin fluorophores as a stable electron donor group and regulate the photophysical properties of these dyes.

BODIPY dyes tend to assemble into *H*-aggregates because of the large π-conjugated frameworks, and most BODIPY dyes usually show ACQ in the aggregation state. In order to solve this problem, BODIPY dyes can be employed as *J*-aggregation scaffolds. However, finding suitable building blocks for BODIPY *J*-aggregates remains a great challenge. In 2009, Meállet-Renault et al. [34] reported on a new type of hindered BODIPY dye prepared by modification of its pyrrole substituents. They introduced a PCP skeleton into the core of BODIPY to increase steric hindrance and prevent π–π stacking, which improved the photophysical properties of the BODIPY; these PCP-BODIPY dyes 6a−d showed red-shifted absorption and emission bands compared with Ph-BODIPY dye 7 (Table 1, entries 10,11, Figure 2). Accordingly, the Stokes shifts of these PCP-BODIPY dyes 6a−d ranged between 881 and 1290 cm^−1^, which were higher values than for the Ph-BODIPY dye 7. These results may be due to the electron-donating nature and the steric hindrance of PCP, which enhance the conjugation effect and limit the possibilities for reorientation of the PCP group in the excited state. In particular, when their films were prepared by drop-casting, their emission bands were narrower than those in solution, and their fluorescence quantum yields increased dramatically and solid-state fluorescence were obtained, which may be attributed to the *J*-like aggregates in the film.

In 2021, Liu and co-workers reported a new PCP-BODIPY dye 8 (Figure 2) [35] as an example of a BODIPY dye with *J*-aggregation, which can induce second near-infrared fluorescence. First, the authors revealed that when the *meso*-position of BODIPY was changed from phenyl to [2.2]paracyclophane, compound PCP-BODIPY dye 8 showed red-shifted absorption and emission bands centered at 722 and 795 nm in comparison with its analog compound Ph-BODIPY 9 (*λ*_abs_ = 700 nm and *λ*_em_ = 750 nm) (Table 1, entries 12,13). This phenomenon can be attributed to the stronger electron-conjugation effect of the PCP group. In addition, they explored the *J*-aggregation behavior of PCP-BODIPY dye 8 by investigating the emission changes in tetrahydrofuran (THF)–water binary solvents with variations in water volumetric fraction. The results indicated that PCP-BODIPY dye 8 can exhibit *J*-aggregation and showed a weak NIR-II emission band around 1000 nm when the water volumetric fractions reached 80% and 90%. Moreover, the authors found that the photophysical properties of Ph-BODIPY 9 indicated *H*-aggregation-induced emission quenching in the aggregation state, which demonstrated that the introduction of the PCP group played a key role in the *J*-aggregation behavior of PCP-BODIPY dye 8.

Because of the NIR-II emission capability of PCP-BODIPY dye 8 in both THF–water and solid state, the authors also investigated the stability of the NIR-II emissive *J*-aggregates in NPs (Figure 2). The results indicated that the NIR-II emissive *J*-aggregates could be efficiently stabilized in a pluronic F-127 matrix. To verify the biological imaging capability of PCP-BODIPY 8, these authors prepared the PCP-BODIPY NPs by encapsulating the PCP- PCP-BODIPY 8 aggregates into a pluronic F-127 matrix and performed both in vitro and in vivo NIR-II imaging. The results of in vitro imaging revealed that the PCP-BODIPY 8 NPs had good NIR-II imaging penetration depth up to 8 mm, which was higher than the 6 mm of the clinically approved NIR-I dye indocyanine green (ICG). Moreover, the results of the in vivo imaging experiment showed that the brightness and clarity of PCP-BODIPY 8 NPs were higher than that of ICG under the same imaging conditions. Accordingly, the fluorescence of PCP-BODIPY 8 NPs decreased gradually with increase in time from 5 min to 24 h, in contrast to the undetectable fluorescence of ICG when the time was increased to 8 h. These results demonstrate that the PCP-BODIPY 8 NPs have higher resolution and longer-term NIR-II imaging ability than ICG. Furthermore, the potential application of PCP-BODIPY 8 NPs in mapping lymph nodes and image-guided cancer surgery was also investigated in nude mice. All the above results indicate that the steric and conjugation effect of the PCP skeleton plays a key role in manipulating the photophysical properties of BODIPY dyes and promoting their application in biological sensing and imaging.

### 2.2. Application of PCP Skeleton in Detection

Of the various Hg^2+^ fluorescent probes, rhodamine-based chemodosimeters for Hg^2+^ have become increasingly attractive because of the rapidity, high sensitivity, and excellent selectivity of their mercury-promoted desulfurization reactions. However, to our best knowledge, the spectroscopic properties of the PCP skeleton in rhodamine-based fluorescent probes have yet not been reported. In 2017, our group reported two novel fluorescent probes 10a,b (Figure 3) [36], which were the first PCP skeleton fluorescent probes based on rhodamine dye. Firstly, we studied the sensing behaviors of 10a,b in different environments; the probe 10a exhibited highly selective and sensitive responses to Hg^2+^ in aqueous solution. The sensing behaviors toward Hg^2+^ were studied by UV–vis and fluorescence spectroscopy in an HEPES buffer (HEPES/EtOH = 1:1 *v/v*, pH 7.2). We found that the gradual addition of Hg^2+^ (from 5 to 90 μM) to the solution of compound 10a resulted in enhanced absorption and fluorescence intensities with a color change from colorless to pink. The results indicate that the probe 10a can serve as a “naked eye” probe for Hg^2+^. We also investigated the pH effect and the long-term response time on the fluorescence intensity with the addition of Hg^2+^. The fluorescent response was stable over a wide pH range of 6−10, and the time-dependent change in absorbance and fluorescence intensity terminated within 15 min. Meanwhile, we also investigated the addition of other ions, such as Al^3+^, Fe^3+^, Ni^2+^, Co^2+^, Cd^2+^, Ca^2+^, Mn^2+^, Pb^2+^, Mg^2+^, K^+^, Na^+^, and Ag^+^, which showed little interference with the fluorescence response resulting from the addition of Hg^2+^. The sensing mechanism was also proposed as being based on the stoichiometric and irreversible Hg^2+^-promoted reaction of the spirolactam 10a to a ring-opened amide. These results demonstrate that compound 10a can be used as a selective and rapid fluorescent probe for Hg^2+^ over various other metal ions. Furthermore, these spectroscopic properties of the through-space delocalized PCP core are analogous to the alkyl and can help researchers to gain insight into the spectroscopic properties of PCP in fluorescent probes. In addition, we evaluated the cell viability of 10a by MTT assay. When the concentration of 10a increased from 2.5 to 10 μM, the percentage viability of the studied A549 cells displayed no significant reduction, which indicates that the fluorescent probe 10a has good biocompatibility in A549 cells. Finally, we also evaluated the capacity of fluorescent probe 10a to operate within living cells. The 10a-stained human A549 lung adenocarcinoma cells were tested before and after the addition of Hg^2+^ by confocal microscopy. By comparing with the MitoTracker Green FM-stained A549 cell, the probe 10a showed cell membrane permeable capability and can be applied in detecting intracellular Hg^2+^ in human lung adenocarcinoma cells. Therefore, the introduction of the PCP skeleton to rhodamine-based chemodosimeters can improve the selectivity, sensitivity, and biocompatibility of rhodamine dyes.

Use of the organic π-conjugated molecules with donor (D)−accept (A) electronic structure is considered an effective strategy for the development of multi-stimuli responsive materials [39]. The reason may be that, when trace water or acidic gases interact with D−A-type organic π-conjugated molecules, the formed hydrogen bonding interaction or protonation [40] can modulate intramolecular electron transfer (ICT), thus changing the emission intensity and color. Because of the π-rich electronic structure of the PCP skeleton [41], it can be used as the electron-donating part in π-conjugated D−A organic materials. In 2022, our group reported a series of D−A-type [2.2]paracyclophanyl 4*H*-pyran-4-one derivatives 11a−d (Figure 3) [37]. Because of its special AIE properties, with increased fluorescent intensity and red-shifted wavelength, we applied compound 11c to detect trace water in THF, 1,4-dioxane, and acetone (Figure 3). The results indicated that the detection limits were much lower than for previous results, with 3.55 ppm in THF, 1.39 ppm in 1,4-dioxane, and 8.75 ppm in acetone. Therefore, 11c can be applied in the high sensitivity detection of trace water in organic solvents. We also studied the relationships between different pH values (2−12) and the emission intensities of 11a−d. Based on these results, only compound 11b exhibited a good linear relationship in pH range from 2 to 7. Thus, 11b can be applied as a water-soluble fluorescent ratiometric pH sensor in the pH range of 2–7. Finally, we investigated the abilities of 11a–d-coated test strips to capture and detect acidic gases (HCl, TFA, HCOOH, SO_2_) and HCHO. Interestingly, because of the different D−A properties, compounds 11a–d exhibited different sensing behaviors and have different potential applications in data encryption and decryption and anti-counterfeiting fields. Due to the similar fluorescent properties in the solid state, 11a and 11b both exhibited enhanced fluorescent intensities and color changes upon being exposed to acidic gases and HCHO vapor. Compound 11d was able to detect acidic gases but showed no obvious change when exposed to HCHO vapor. Importantly, the test strips of 11c could detect acidic gases with significantly quenched emission intensities and obvious color change. In particular, the emission intensities and original color of the test strips of 11a, 11c,d could be recovered when the adsorbed test strips were continually exposed to NH_3_ vapor. These sensing mechanisms were all elucidated by X-ray diffraction analysis, NMR, and time-dependent density functional theory (TD-DFT) calculations. The mechanisms of the trace water and HCHO detection were based on hydrogen bond interactions, and the mechanisms of the pH values and acidic gases detection were based on the protonation and deprotonation of the carbonyl, methoxy, and *N,N*-dimethylamino groups. Accordingly, the potential applications of 11a–d in the fields of information recording and security technologies were also demonstrated. Therefore, because of the electron-conjugating effect of the PCP skeleton on the pyranone ring, a new family of π-conjugated D−A-type sensors 11a–d was constructed, which exhibited higher sensitivity in detecting trace water, pH values, acidic gases, and HCHO.

### 2.3. Application of PCP Skeleton in CPL Modulation

Due to the stable planar chirality [42], π-conjugated effect [43], and steric effect [44] of the PCP skeleton, it can be used as an ideal scaffold to construct an AIE-active CPL emitter. In 2021, our group reported a new family of CPL-active BF_2_ complexes by fusing a *N^O*-chelated BF_2_ complex with a planar chiral PCP skeleton to give (*R*_p_)/(*S*_p_)-12a−e (Figure 4) [38]. The molecular structure and the photophysical behaviors of (*R*_p_)/(*S*_p_)-12a−e were investigated by X-ray diffraction and TD-DFT calculations, respectively. The results confirmed the efficient steric effect, intermolecular interactions, and intramolecular charge transfer (ICT) effect of the PCP group. We then investigated the chiroptical properties of these planar chiral BF_2_ complexes (*R*_p_)/(*S*_p_)-12a−e by CD and CPL spectroscopy in DCM. In the CD spectra, (*R*_p_)/(*S*_p_)-12a−e exhibited intense and mirror-image Cotton effects with |*g*_abs_| values of approximately 5.6 × 10^−3^, 6.2 × 10^−3^, 1.9 × 10^−3^, 3.1 × 10^−3^, and 4.6 × 10^−3^, respectively. Moreover, (*R*_p_)/(*S*_p_)-12a−d also exhibited intense and mirror image CPL signals with maximum |*g*_lum_| values of 4.5 × 10^−3^, 6.2 × 10^−3^, 5.4 × 10^−3^, and 6.2 × 10^−3^, respectively. These results were comparable to the typical |*g*_lum_| values of the CPL-active PCP-fused small organic molecules in solutions reported by Chen et al. [45,46,47]. Since these BF_2_ complexes exhibited aggregation-induced emission enhancement (AIEE) properties in the mixed solvent systems of THF/H_2_O (Table 1, entry 12), their chiroptical properties in aggregated state were also investigated. Apparently, the CD signals of (*R*_p_)/(*S*_p_)-12a−e suddenly collapsed at 300−400 nm, and an additional CD band appeared at around 400−500 nm, which indicated that chiral aggregates were formed. Importantly, the CPL spectra of (*R*_p_)/(*S*_p_)-12a−e exhibited aggregation-amplified CPL with larger |*g*_lum_| values of 6.2 × 10^−3^, 7.1 × 10^−3^, 7.6 × 10^−3^, 5.3 × 10^−3^, and 6.1 × 10^−3^ for (*R*_p_)/(*S*_p_)-12a−e in the aggregated state, respectively. Furthermore, the CPL properties of (*R*_p_)/(*S*_p_)-12a−e in the solid state were also investigated with the |*g*_lum_| values found to be 5.4 × 10^−4^, 1.8 × 10^−3^, 4.5 × 10^−4^, 1.9 × 10^−3^, and 7.7 × 10^−4^ for (*R*_p_)/(*S*_p_)-12a−e, respectively. These results indicate that efficient CPL materials with moderate quantum yields and moderate |*g*_lum_| can be obtained by incorporating a PCP skeleton into a boron difluoride complex.

In 2022, our group reported AIE-active PCP-based aurones (*R*_p_)/(*S*_p_)-13 (Figure 4) [29], which were constructed by fusing an aurone moiety with PCP and exhibited aggregation-induced CPL (AICPL) properties. Interestingly, (*R*_p_)/(*S*_p_)-13 were prepared by a convenient one-step reaction of (*R*_p_)/(*S*_p_)-5-acetyl-4-hydroxy[2.2]paracyclophane with p-nitrobenzaldehyde. This is a special reaction for the preparation of (*R*_p_)/(*S*_p_)-13 only, which may be due to the electron-donating conjugation effect of the PCP group and the electron-withdrawing effect of the nitro group. We also compared the fluorescence properties of (*R*_p_)/(*S*_p_)-13 with the phenyl-based aurone (Figure 4) and found that (*R*_p_)/(*S*_p_)-13 exhibited stronger ICT character and typical AIE behavior. Moreover, (*R*_p_)/(*S*_p_)-13 showed moderate fluorescence with yellow color under UV-irradiation compared with the weak fluorescence and nattier blue color of the phenyl-based aurone. We also compared the crystal structures and the TD-DFT calculation results of (*R*_p_)/(*S*_p_)-13 and the phenyl-based aurone; the results indicated that the substitution of the phenyl ring with a PCP ring in aurone compounds could increase the electron-donating ability and steric hindrance, hindering intermolecular π−π stacking interactions and improving the photophysical properties of the aurone compounds. Furthermore, we investigated the chiroptical properties of (*R*_p_)/(*S*_p_)-13 in THF/H_2_O with varying volumetric fractions of water (*f*_w_). The CPL signals firstly decreased as *f*_w_ was increased from 0% to 70% and then gradually increased as *f*_w_ increased from 70% to 90%, which demonstrated the AICPL effect of (*R*_p_)/(*S*_p_)-13. Accordingly, larger *g*_lum_ values were obtained with ±3.7 × 10^−3^ at 578 nm in the aggregated state. Therefore, the introduction of a planar chiral PCP skeleton to the aurones increased the electron-donating ability and steric hindrance, which resulted in the AIE effect and led to efficient AICPL-active aurones with moderate *Φ*_f_ and high |*g*_lum_| values (Table 1, entry 13).

In 2023, our group also reported a series of PCP-based 4-(dicyanomethylene)-4*H*-pyran compounds (*R*_p_)/(*S*_p_)-14a−d by introducing a dicyanovinyl group to modify the planar 4*H*-pyran-4-one compounds 11 (Figure 4) [30]. We also investigated the X-ray structures, TD-DFT calculations, electrochemical properties, and photophysical properties in solution and solid state. The results demonstrated that these compounds exhibited AIE properties at *f*_w_ = 99% and showed intense photoluminescence, with the color changing from orange to deep red with high *Φ*_f_ of up to 97% in solid state. These results indicated that the introduction of PCP and a dicyanovinyl group could hinder intermolecular π–π stacking interactions and enhance the ICT process, which led to the high solid-state *Φ*_f_ and deep red color. We then investigated the CD spectra of (*R*_p_)/(*S*_p_)-14a,b in THF/H_2_O mixture with different *f*_w_. The results showed that the CD signals were both enhanced when *f*_w_ was increased to 90%. Meanwhile, we investigated the CPL behaviors of (*R*_p_)/(*S*_p_)-14a,b in aggregated and solid states. Interestingly, (*R*_p_)/(*S*_p_)-14a,b were CPL silent either in THF solutions or in aggregated states, which may have been due to the weak emission of (*R*_p_)/(*S*_p_)-14b and the low-degree oligomeric stacking of (*R*_p_)/(*S*_p_)-14a in the aggregated state determined by the FE-SEM images. However, (*R*_p_)/(*S*_p_)-14a,b exhibited intense CPL signals in a solid state, and the signals were centered at 662 and 617 nm with |*g*_lum_| values of 1.9 × 10^−3^ and 1.8 × 10^−3^, respectively. In comparison with most reported PCP derivatives, (*R*_p_)/(*S*_p_)-14a exhibited a longer CPL maximum wavelength and higher *Φ*_f_. These results suggest that the unique structure of the planar chiral PCP skeleton and the strong ICT process can be utilized in preparing red AICPL materials.

## 3. Conclusions

In this review, we summarized the different applications of the PCP skeleton in the fields of dye modification, fluorescent probes in detection, and CPL modulation. Firstly, regarding dye modification, we demonstrated the PCP skeleton could provide a 3D structure for coumarin systems, and that these coumarins showed extremely large Stokes shifts (up to 230 nm) and red-shifted absorption and emission bands. Furthermore, the analysis of BODIPY dye (PCP-BODIPY dyes 6, 8) revealed that the introduction of a PCP group could increase steric hindrance and prevent π–π stacking, thus improving the fluorescence properties of dyes and inducing *J*-aggregation. Because of the conjugation effect and steric effect of the PCP group, PCP-BODIPY dye 8 showed both NIR-I (*J*_1_-band, 900 nm) and NIR-II (*J*_2_-band, 1010 nm) emission in THF–water binary solvent and could be used for lymph node imaging and fluorescence-guided surgery in nude mice.

Second, because of the π-rich character of the PCP moiety, it can be used as the electron-donating component in π-conjugated D−A structures and improve the spectroscopic properties of fluorescent sensors. Accordingly, the PCP-rhodamine dye 10a can be used as a selective and rapid fluorescent probe for Hg^2+^; PCP-4*H*-pyran-4-one 11c facilitates the visual and quantitative detection of trace water in organic solvents with low detection limits of 1.39–8.75 ppm; PCP-4*H*-pyran-4-one 11b exhibits fluorescence ratiometric detection of pH in the range of 2–7; PCP-4*H*-pyran-4-one 11a−d could serve in coated test strips for detecting acidic gases and HCHO with different sensing properties. Hence, these PCP-based fluorescent sensors exhibit highly sensitive multi-stimuli response behaviors and have potential in intracellular Hg^2+^ detection, data encryption and decryption, and anti-counterfeiting applications.

Finally, the planar chiral PCP is an ideal scaffold to construct excellent CPL emitters. Following incorporation of the PCP skeleton, the resulting boron difluoride complexes, aurone compounds, and 4-(dicyanomethylene)-4*H*-pyran compounds all exhibited aggregation-amplified CPL with moderate |*g*_lum_| values up to 7.6 × 10^−3^. Moreover, because of the stable planar chirality and excellent fluorescence properties, the application of these PCP-based dyes to the development of planar fluorescent probes is underway in our laboratory and will be reported in due course. We anticipate that this review may provide readers with a comprehensive understanding of the PCP skeleton, prompting further improvements in fluorescent dye design and the exploration of wider applications of PCP derivatives.

## Data Availability

Not applicable.

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
