# Peer review of "Recent Progress in Research on [2.2]Paracyclophane-Based Dyes"

_molecules, 2023, doi:10.3390/molecules28072891_

Round 1
Reviewer 1 Report
In this review, Duan et al. summarized and emphasized the development of [2.2]paracyclophane-based dyes. Because of the unique structure of [2.2]paracyclophane, which provides planar chirality, electron-donating capacity, and steric hindrance, it is widely used in functional organic materials such as emitter and CPL materials. This review is highly fascinating, well-organized, and it will be valuable for the development of multifunctional dyes based on [2.2]paracyclophane due to its peculiar planar chiral structure and unique emissive property. I therefore advise publishing this review in Molecules after minor revisions.
1. Some related references should be provided, such as the references for the asymmetric synthesis, π-stacked polymers, energy materials, and organic fluorescent dyes of [2.2]paracyclophane skeleton.
2. There are small errors in different parts of the manuscript.
Example:
"These skeleton have twisted structures” should be “These skeletons have twisted structures”;
"the introduction of PCP skeleton to the organic fluorescent dyes improve the photophysical properties” should be “the introduction of PCP skeleton to the organic fluorescent dyes improves the photophysical properties”;
Author Response
Comments:
In this review, Duan et al. summarized and emphasized the development of [2.2]paracyclophane-based dyes. Because of the unique structure of [2.2]paracyclophane, which provides planar chirality, electron-donating capacity, and steric hindrance, it is widely used in functional organic materials such as emitter and CPL materials. This review is highly fascinating, well-organized, and it will be valuable for the development of multifunctional dyes based on [2.2]paracyclophane due to its peculiar planar chiral structure and unique emissive property. I therefore advise publishing this review in Molecules after minor revisions.
Response:
Thanks for the positive comments and we appreciate that! We have revised the manuscript according to your comments.
Question 1 (Q1): Some related references should be provided, such as the references for the asymmetric synthesis, π-stacked polymers, energy materials, and organic fluorescent dyes of [2.2]paracyclophane skeleton.
Answer 1 (A1): The related references have been added as references 5−8 in the main text.
- Liang, H.; Guo, W.C.; Li, J.X.; Jiang, J.J.; Wang, J. Chiral Arene Ligand as Stereocontroller for Asymmetric C−H Activation. Chem. Int. Ed. 2022, 61, e202204926.
- Morisaki, Y.; Chujo, Y. Synthesis of π-Stacked Polymers on the Basis of [2.2]Paracyclophane. Chem. Soc. Jpn. 2009, 82, 1070−1082.
- Yu, H.; Arunagiri, L.; Zhang, L.; Huang, J.C.; Ma, W.; Zhang, J.Q.; Yan, H. Transannularly conjugated tetrameric perylene diimide acceptors containing [2.2]paracyclophane for non-fullerene organic solar cells. Mater. Chem. A, 2020, 8, 6501−6509.
- Zhang, M.Y.; Peng, Q.; Zhao, C.H. High-contrast mechanochromic fluorescence from a highly solid-state emissive 2-(dimesitylboryl)phenyl-substituted [2.2]paracyclophane. Mater. Chem. C, 2021, 9, 1740−1745.
Q2: There are small errors in different parts of the manuscript.
Example:
"These skeleton have twisted structures” should be “These skeletons have twisted structures”;
"the introduction of PCP skeleton to the organic fluorescent dyes improve the photophysical properties” should be “the introduction of PCP skeleton to the organic fluorescent dyes improves the photophysical properties”;
A2: Thanks for your kind reminder! We have corrected these errors.
Reviewer 2 Report
In this paper, the authors focused on the skeleton of PCP, and summarized the applications of this unique moiety involved materials in dye modification, fluorescent probe, and CPL modulation. The review is short but compact with sufficient information centered on the specific skeleton and its influences to materials. In terms of the art work, I suggest list some figures demonstrating the unique properties or applications of the materials which will help to understand the function of PCP.
Author Response
Thanks for the positive comments and we have added Figures 2−4 to demonstrate the unique properties and the applications of [2.2]paracyclophane materials.
Please see the attachment.

Author Response
Thanks for your comments, we have improved the written quality through the aid of a fluent English speaker and revised the manuscript according to your suggestions.
Please see the attachment

Round 2
Reviewer 3 Report
The written quality has substantially improved and I am happy for this to be published.